# Validation of a Food Knowledge Questionnaire on Tanzanian Women of Childbearing Age

**DOI:** 10.3390/nu14030691

**Published:** 2022-02-07

**Authors:** Maria Vittoria Conti, Marco Gnesi, Rachele De Giuseppe, Francesca Giampieri, Maria Cristina Monti, Naelijwa Mshanga, Joyce Kinabo, John Msuya, Hellas Cena

**Affiliations:** 1Laboratory of Dietetics and Clinical Nutrition, Department of Public Health, Experimental and Forensic Medicine, University of Pavia, Via Bassi 21, 27100 Pavia, Italy; mariavittoria16.conti@gmail.com (M.V.C.); hellas.cena@unipv.it (H.C.); 2Section of Biostatistics and Clinical Epidemiology, Department of Public Health, Experimental and Forensic Medicine, University of Pavia, 27100 Pavia, Italy; marco.gnesi@unipv.it (M.G.); cristina.monti@unipv.it (M.C.M.); 3Department of Biochemistry, Faculty of Sciences, King Abdulaziz University, Jeddah 21589, Saudi Arabia; 4Research Group on Food, Nutritional Biochemistry and Health, Universidad Europea del Atlantico, 39011 Santander, Spain; 5The Nelson Mandela African Institution of Science and Technology, Arusha 447, Tanzania; mshanganaelijwa@gmail.com; 6Department of Food Technology, Nutrition and Consumer Sciences, Sokoine University of Agriculture, Morogoro 3000, Tanzania; joyce_kinabo@yahoo.com (J.K.); j_msuya@yahoo.com (J.M.); 7Clinical Nutrition and Dietetics Service, Unit of Internal Medicine and Endocrinology, ICS Maugeri IRCCS, 27100 Pavia, Italy

**Keywords:** food knowledge, awareness, childbearing age women, Tanzania, Sub-Saharan Africa, nutrition, health promotion, disease prevention

## Abstract

The present manuscript describes the validation of a food knowledge questionnaire (FKQ) for Tanzanian childbearing age women. The FKQ was derived from the Ugandan version and was adapted according to Tanzanian culture and food habits, including 114 closed-ended questions, divided into five different sections. The FKQ was administered to 671 Tanzanian childbearing-age women who were eligible if they: (i) were aged between 14 and 49 years old, (ii) had not been diagnosed with any disease and (iii) lived in the urban or peri-urban area of the Arusha and Morogoro region. The validation process of the FKQ was conducted in Tanzania and the recruitment occurred between August and October 2020. The final version of the validated questionnaire was characterized by a total of 88 questions, divided into ten different sections; each section aimed to investigate a different aspect of food knowledge, except for section A, which collected information related to the social and demographic characteristics of the respondent. The food knowledge questionnaire showed good construct validity and content validity to assess knowledge and food practices in Tanzanian women of childbearing age and could be used in future studies to identify women at higher risk of unhealthy eating habits and food choices.

## 1. Introduction

The term “food knowledge” (FK) refers to “the competence to understand healthy nutrition concepts” [1] or “the understanding of basic facts about food and nutrition” [2,3]. Although a unique definition for the FK concept is lacking, it has been demonstrated that FK impacts individuals’ lives due to its relationship with food behavior and eating habits [1]. For instance, data from cross-sectional studies of large cohorts demonstrated how FK can affect diet quality, with high FK scores associated with increased consumption of fruit and vegetables [1].

Food and nutrition knowledge has become particularly important in low-income countries, including Sub-Saharan Africa [2], where numerous findings have shown that a lack of FK contributes to malnutrition [2]. This association becomes even more dramatic in vulnerable population groups, including women of childbearing age (CBA), who have increased nutritional requirements for appropriate fetal programming and development, besides their health [4,5]. Inadequate FK in CBA women places their future and that of their offspring at risk of nutrition-related chronic diseases, also feeding the intergenerational cycle of malnutrition and poverty [6]. It has been reported that adolescent girls’ offspring have approximately a 40% chance to survive beyond their fifth birthday if they have adequate food knowledge [6,7]. Therefore, ensuring adequate FK among women of CBA is a key strategy to assure health and well-being, not only for themselves but also for the whole community. Health promotion may indeed be achieved directly and indirectly, since women are primary caregivers and impact their children’s nutrition throughout their nutritional status as well as using childcare practices [8]. Furthermore, women are mostly in charge of food purchases, cooking and meal planning for their family, and sometimes for part of the village [6,9]. Therefore, specific actions should be integrated into countries’ strategies to avoid unpleasant repercussions for their nutritional status as well as for the growth and development of their children and, overall, the future human capital [10,11].

In this context, current data highlight the need to develop easy-to-use, population-specific tools for FK assessment, but, thus far, evidence of their validity (and particularly of construct validity) is scarce [10]. In the last few decades, Parmenter and colleagues developed for the first time an FK questionnaire (FKQ), tailored to the adult UK population, to identify areas of weakness in people’s knowledge of healthy eating [12]. Similarly, from the pioneering work of Parmenter and colleagues, other researchers adopted the questionnaire in different populations. Recently, Bukenya et al. developed a questionnaire for the Ugandan population, investigating FK of food groups, food choice, the relationship between nutrition and disease and finally food fortification [13].

The present manuscript, starting from the questionnaire of Bukenya et al. [13] and driven by the need to draft a specific screening tool for the Tanzanian population, describes the FKQ’s validation for Tanzanian women of CBA.

## 2. Materials and Methods

### 2.1. Questionnaire Design

The FKQ that underwent validation in the present study was derived from the Ugandan version [13] and then adapted according to Tanzanian culture and food habits, as previously reported elsewhere [9]. The Tanzanian translation and cultural adaptation of the FKQ [9] included 114 closed-ended questions and was divided into five different sections. In particular, section A (17 questions) aimed at collecting the socio-demographic and economic characteristics of the respondent; section B (5 questions) was related to “what people think about nutrition expert advice on healthy food habits”; section C1 (7 questions) was related to “knowledge about the classification of foods groups”; section C2 (7 questions) was related to “knowledge about macro-/micro-nutrients content in different foods” and section D (6 questions) was related to “knowledge on the impact of nutrition on health” (9).

Items were written in the form of a “test”, meaning that each item was a closed-ended question with only one correct answer [9]. For each item, answers were coded as “correct” or “not correct”. As shown in a previous exploratory analysis [9], these items were able to discriminate between those with “correct food knowledge” and those with “incorrect food knowledge” archetypes.

### 2.2. Study Setting for the Validation

The validation process of the present FKQ was conducted in Tanzania. The country is divided into 31 administrative regions, 21 in mainland Tanzania and 5 in the Archipelago of Zanzibar. Regions are divided into administrative district councils, which are in turn divided into wards. The study area was located in the Arusha region, whose regional capital is the city of Arusha (the third major urban center of the country), and in the Morogoro region.

The recruitment was conducted both in urban areas (city of Arusha and Morogoro) and in peri-urban ones (defined as areas of transition from strictly rural to urban), in places where volunteers could gather, before the COVID-19 pandemic. Mkundi Ward, located 77 Km west of Morogoro Municipality along the Morogoro Dodoma highway, was selected purposively for the study out of the 20 wards, as it fitted well with the description of the requirements of the study. Six streets were sampled randomly from 15 streets.

### 2.3. Sampling

According to the rule of thumb of enrolling 4–10 subjects per item, and given the 88 items of the questionnaire (excluding those in section A), the sample size of 671 CBA women (corresponding to 7.6 subjects per item) was deemed adequate. All women were considered eligible if (i) they were 14–49 years old; (ii) they had not been diagnosed with any disease and (iii) they lived in the urban or peri-urban area of the Arusha and Morogoro region. The recruitment occurred between August and October 2020. All eligible women were invited to participate in meetings organized in the different wards. The questionnaire, written in English, was translated into Swahili using the forward-backward translation method according to the WHO guidelines [14].

Study approval was granted by the Sokoyne University of Agriculture (SUA), located in Morogoro Municipality (Tanzania). The Vice-Chancellor of the SUA was empowered to issue research clearance to staff, students, research associates and researchers of SUA on behalf of the Tanzania Commission for Science and Technology (Registration number: DPRTC/R/142 Val. II/10) within the Sustainable Agri-Food Systems Strategies project, funded by the Italian Ministry of University and Research (MIUR) (Fondo integrativo speciale per la ricerca, FISR; CUP: H42F16002450001). Moreover, data were collected in close collaboration with the Arusha City Council and Morogoro Municipal Council. Each respondent of legal age signed written informed consent, while the informed written consent of a parent or legal guardian was required for subjects aged <18.

### 2.4. Questionnaire Administration

The FKQ that underwent the validation process was administered as an interview in the Swahili language by a trained enumerator, who had been instructed on how to administer questionnaires, of Sokoine University of Agriculture and Istituto OIKOS East Africa, a Tanzanian NGO based in the city of Arusha, where it promotes the protection of biodiversity and the sustainable use of natural resources to fight poverty [15].

All the answers to the FKQ were collected using the Open Data Kit (ODK) application [16], an open-source mobile data software program to collect data quickly, accurately, offline and at scale. The data collected were kept in cloud storage and then transferred for statistical analysis. Administration of the original 88-item questionnaire by a trained enumerator took 15 min per person.

### 2.5. Descriptive Statistics

All statistical analyses were performed using Stata 13.1 [17]. Data were described as mean and standard deviation (SD) for continuous variables, or median and interquartile range (IQR) for ordinal variables and continuous variables with non-symmetrical distributions; categorical variables, as well as items’ responses, were described as frequencies and percentages. Descriptive analysis of items’ scores helped to identify those items where a relevant share of respondents (approximately >90%) gave a correct (or wrong) answer; such items, being too easy or too difficult, may show low discriminating power in discerning subjects depending on their level of FK [18].

### 2.6. Construct Validity

Section A was excluded from construct validation since it was only aimed at collecting socio-demographic data rather than measuring FK. Therefore, validation comprised B-L sections.

Given the high number of items and sections, the factorial structure was ascertained with a multi-step approach. At each step, Exploratory Factor Analysis (EFA) was used; factors were extracted by applying the Principal Component Factor (PCF) method on the matrix of tetrachoric correlations between items. For a better interpretation, factorial solutions were rotated through an orthogonal rotation (Quartimax method). First, EFA was applied to assess the factorial structure of each section separately. Then, for all the items that did not fit within the identified factors, it was assessed whether they could be semantically coherent with other factors outside their original section; they were added to these factors, and such new factors were inspected using another EFA. Finally, all the items that were not yet included in any of the identified factors were analyzed together in a new EFA to identify any additional unpredicted factors.

Each factor of the final structure was analyzed again with EFA to confirm its unidimensionality and estimate uniqueness. Each factor was also inspected with Multiple Correspondence Analysis (MCA) (Burt’s matrix method) to better assess the properties of the scoring system of the factor’s items; this step helped in the identification of issues with the scorings (e.g., when the score of the correct answer to a certain item was not consistent with the scores of the correct answers of other items in the same factor, for instance, because it had a negative score value while the other correct answers had a positive score value). The factors, as they were identified at the end of such a process, corresponded to the final sections of the FKQ.

For each factor, a total score was computed as the sum of items’ scores. Factors’ scores were described, and their correlations were inspected using Spearman’s correlation coefficient (correlation matrix not shown). A total FKQ score was computed by summing the scores of each section (section A excluded).

To assess the existence of a potential super-factor, the scores of all the factors were included in a Structural Equation Model (maximum likelihood method, robust estimator of the variance–covariance matrix, standardized coefficients).

### 2.7. Content Validity

Factor scores were tested against some socio-demographic variables of interest, including educational level and nutrition-related qualification, to assess how the Tanzanian FKQ responded to changes in variables that might have been related to women’s underlying FK. Differences in scores between groups were tested by using Mann–Whitney’s test (for two groups) or the Kruskal–Wallis test (for more than two groups, followed by Mann–Whitney’s as post hoc tests if applicable). The significance threshold was set at 5% (α = 0.05); when applicable, Bonferroni’s correction was applied to adjust for multiple testing.

## 3. Results

The sample consisted of 671 women (mean age of 33.9 years; SD 10.9 years); a general description of the sample (section A) is reported in Table 1.

In brief, most of the women had primary or secondary education (49.3% and 29.8%, respectively), and two thirds were workers (66.0%). Moreover, 70.3% of women were married, with at least one son or daughter (90% of cases). Only 13.6% of participants were on a diet.

A description of their responses is reported in Table 2.

In detail, fourteen of the 88 items yielded 90% or more correct answers, 11 of which were dropped because they were not well integrated into the final factorial structure. In MCA, 16 items showed potential issues with the scoring system, which were not consistent with those of other items; only one such item was not dropped from the final structure, but its scoring inconsistency was indeed borderline. One item was characterized by both the previous conditions and was dropped from the final structure. EFA led to the dropping of a further 18 items that were not integrated into the factorial structures of the sections.

Besides socio-demographic information items (section A), the final structure of the questionnaire (Appendix A) consisted of nine sections, including a total of 43 items investigating the content of sugars, protein, fiber, sodium and fat, besides knowledge of nutrition-related diseases. The EFA results of the final structure are reported in Table 3. All factors were unidimensional.

*Section B:* This section aimed at investigating knowledge of how to maintain good health according to experts’ recommendations, using specific questions on adequate daily intake of fruit and vegetables, on the role of different food items containing lipids and on processed food consumption as well as food sources with high fiber content. The factor of section B explained 48.1% of the total observed variance. Its six items had moderate factor loadings (ranging from 0.61 to 0.78) and uniqueness. As shown by MCA (Appendix A), the items’ scoring was consistent (i.e., going in the same direction from correct to wrong) and substantially homogeneous (i.e., showing comparable values) across all items.

*Section C:* This section investigated the knowledge of sugar content in foods generally consumed every day. The factor of section C included three items, with relatively homogeneous factor loadings ranging from moderate to high (0.68–0.81), moderate uniqueness and 55.2% of explained variance. Scorings, as shown by MCA (Appendix A), were consistent, even though they were disclosed to be less homogeneous than in the previous factor.

*Section D*: Section D explored the knowledge of protein content in foods habitually consumed every day. The factor of section D explained 61.5% of the total observed variance of its items and included four items; factor loadings were high (>0.82) for all items except one, which also had higher uniqueness. Nonetheless, MCA (Appendix A) disclosed a consistent and homogeneous scoring.

*Section E:* This section investigated the sample’s knowledge about dietary fiber. The factor of section E explained 69.6% of the total variance, and its four items showed very high factor loadings (0.82–0.910), with a single exception (0.73), and optimal uniqueness. MCA (Appendix A) showed consistent and homogeneous scoring.

*Section F:* Section F evaluated the knowledge regarding sodium content in food, as well as health conditions related to high sodium intake. The factor of section F explained 50.5% of the observed variance and included four items, with moderate factor loadings (0.61–0.81) and uniqueness. MCA (Appendix A) showed consistent and homogeneous scoring.

*Section G:* Section G was related to food group classification. It was composed of questions on starchy food and healthy alternatives to red meat consumption. The factor of section G included six items and explained 61.4% of the total variance. Factor loadings were high and substantially homogeneous (0.72–0.84), and uniqueness was essentially optimal. Scoring was consistent in MCA (Appendix A), although little inhomogeneity was observed (particularly between the values of a wrong answer).

*Section H:* Section H was related to dietary composition and enquired about whether some food items are high in fat or fiber. The factor of section H explained 54.8% of the total observed variance of its three items. Factor loadings were moderate or high (0.63–0.83), with moderate uniqueness. Similar to the previous factor, MCA (Appendix A) showed consistent scoring with little inhomogeneity, particularly in the value of a wrong answer.

*Section I*: Section I was composed of three questions and investigated food choice appropriateness considering the dietary composition of food items. The factor of section I was the least robust in psychometric terms: it explained 45.6% of the observed variance of its three items, and one item had a quite low factor loading (0.49, against 0.75 of the other two items). Uniqueness was also higher. In MCA (Appendix A), the scoring, albeit consistent, was less homogeneous.

*Section L*: Section L focused on the knowledge of the relationship between nutrition and diseases. The factor of section L was the largest one, including 10 items; its explained variance was 59.1%. Factor loadings were not very homogeneous but were moderate or high (0.66–0.88); uniqueness was moderate. In the MCA (Appendix A), scoring showed consistency and fair homogeneity.

### Section Scores and Total FKQ Scores

Sections’ scores were then computed by summing up the scores of each section’s items. A summative score for each factor was justified by the substantial homogeneity of scores in the MCAs of the factors. In all sections, the scores covered the entire range of theoretically possible values, were characterized by asymmetrical distributions and showed relevant variability (Figure 1).

The scores were left-skewed for several factors (B, C, F, I), with median values one or two points below the maximum. Factors D and L had a bimodal distribution in which both lower and higher scores were well represented. For factors E, G and H, the distribution showed a substantial preponderance of the highest values over the lowest ones.

SEM was applied to factors’ scores to assess the relationship between factors, particularly the existence of a super-factor connecting all the factors. In the hypothesized structure, the super-factor was loading all the factors. Such a model showed an acceptable coefficient of determination (0.84), an almost acceptable RMSEA (0.08) and SRMR (0.051) and a sub-optimal Tucker–Lewis index (0.88).

The relationship between the super-factor and the factors was relevant for most factors, with three exceptions: factors H, I, L were less loaded by the super-factor. Nonetheless, the interpretation of SEM modification indices did not suggest relevant changes in the structure, and different structures (dropping factor I or modeling two super-factors) did not result in significant improvements in the model’s fit.

Lastly, each factor’s score was tested against other variables. All factors had significantly different scores depending on the women’s education level, showing a trend of higher scores with increasing education levels (Appendix A). On the other hand, no differences were observed when comparing women with or without a nutrition-related qualification, with the only exception of factor B, in which women without such qualifications tended to score higher (Appendix A).

The construction of a total FKQ score was then explored. Such a summative score ranged from 1 to 43 (its theoretical maximum value), with a median value of 30 (IQR 11). The value at the 75th percentile was 35. The distribution of the total FKQ score is represented in Figure 2.

## 4. Discussion

The present study aimed to assess the validity of FKQ to be used in women of childbearing age in Tanzania or in a similar socio-cultural context.

This study presents methodologically sound evidence of the construct validity of the FKQ, often lacking in similar questionnaires, thus supporting the use of this questionnaire to assess FK in Tanzanian women of childbearing age.

This specific population has been chosen since the authors have been working on such a target in a multidisciplinary project (SASS) funded by the Italian Ministry of Research and University. Both women and children represent the most vulnerable populations, being more exposed to unsatisfied nutritional requirements in some social and geographical contexts, thus resulting in a negative impact on health and well-being.

Although Tanzanian society is patriarchal, the concept of women playing a marginal role is overcome: their actual role is central to the food system, impacting the health of families and offspring, both indirectly through nutrition during pregnancy and directly through childcare [19]. The first thousand days of life are considered an important time window for the imprint of future life. Recently, even pre-conception nutrition has been shown to play a key role in fetal programming and future development [20]. Moreover, the strong connection between nutritional status and the health of women and future unborn children is well documented in the literature [19].

Women in Tanzania are the pillar of the food system [8]. They work in the agricultural sector and resell their products in the local markets; they lead the meal planning, food purchasing, preparation and cooking for the whole household, and they also deeply shape and influence their own children’s dietary habits and future food choices [8].

The FKQ is designed for the healthcare professionals (such as medical doctors, dietitians and sociologists but also teachers in schools, in universities, etc.) that face people at risk of under/over-nutrition in their daily practice. It is intended as a screening tool, representing the basis on which targeted and population-specific education interventions will be built to achieve adequate nutrition and enhance health and well-being, raising awareness of the link between our food choices and our health.

Although the prevalence of malnutrition in Tanzania has changed a lot over the past 20 years, it is still recognized as an emergency condition.

The prevalence of the underweight condition in Tanzanian women over 18 years old changed from 15.5% in the 2000s to 11.5% in 2019; the overweight condition increased from 23.4% in the 2000s to 38.1% in 2019; the obesity condition was registered to be 15.2% in 2019 and only 5.9% in the 2000s [21]. Thus, it is undeniable how the development of an easy-to-use screening tool can be of support for healthcare professionals to counteract malnutrition in this country [21,22].

The final validated questionnaire consists of different sections, as previously described, with each one focused on a specific aspect of nutrition; the questionnaire allows those who will use it to investigate knowledge about the impact of nutrition on health. Specifically, the questionnaire is aimed at assessing knowledge related to healthy eating habits, including fruit and vegetable intake, as well as sugar, protein, fiber, salt and starchy food consumption. Moreover, SEM highlighted a super-factor including all the FKQ sections and, although not all the sections were as strongly loaded by such a super-factor, this endorses the existence of a common trait beyond the thematic sections of the FKQ, i.e., a “general FK”.

It is interesting to note that the scores of FKQ sections tended to be higher in women with higher levels of education, but scores recorded for women with nutrition-related qualifications tended to be comparable to those of other women.

The questionnaire presented in this manuscript has various strengths. First, to our knowledge, this is the only tool in the scientific literature providing evidence of validity for FK assessment in Tanzanian CBA women. This questionnaire has been found to be an easy-to-use screening tool, quick to complete and, in our opinion, useful in the design of target-specific public health interventions.

Moreover, the large sample (n = 671) recruited for this tool’s validation guarantees the robustness of the results. In addition, during the validation process, some questions were deleted and, with respect to the administered questionnaire, the validated one was shorter—the time required for the FKQ administration was around 10 min.

Some limitations of the present study should also be acknowledged. First of all, water intake is missing, due to its absence in the main structure of the original version to which the authors adhered [13]. Second, the authors are aware that further evidence supporting the validity of the proposed total FKQ score may be needed since this was not the primary objective of the present work. Since the statistical analyses presented here provide limited support for its computation and interpretation, the authors suggest further investigations and recommend a cautious approach to the total score.

## 5. Conclusions

In conclusion, this FKQ showed good construct validity and content validity to assess knowledge and food practices in Tanzanian women of childbearing age. This questionnaire could be used in future studies to identify women at higher risk of unhealthy eating habits and food choices. It could be used also to identify those who would benefit from an educational intervention, which will likely exert a “domino” positive impact on the nutritional status and well-being of the whole society.

## Figures and Tables

**Figure 1 nutrients-14-00691-f001:**
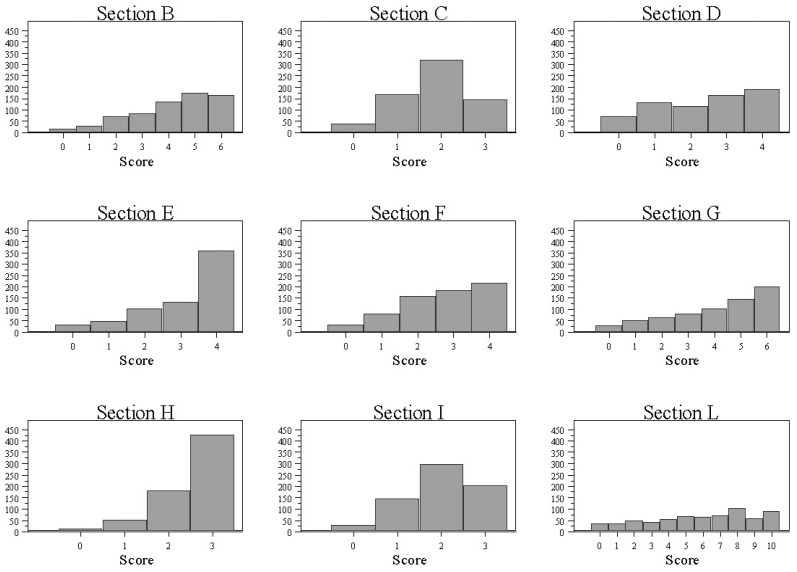
Distribution of sections’ scores. (Vertical axis: frequency).

**Figure 2 nutrients-14-00691-f002:**
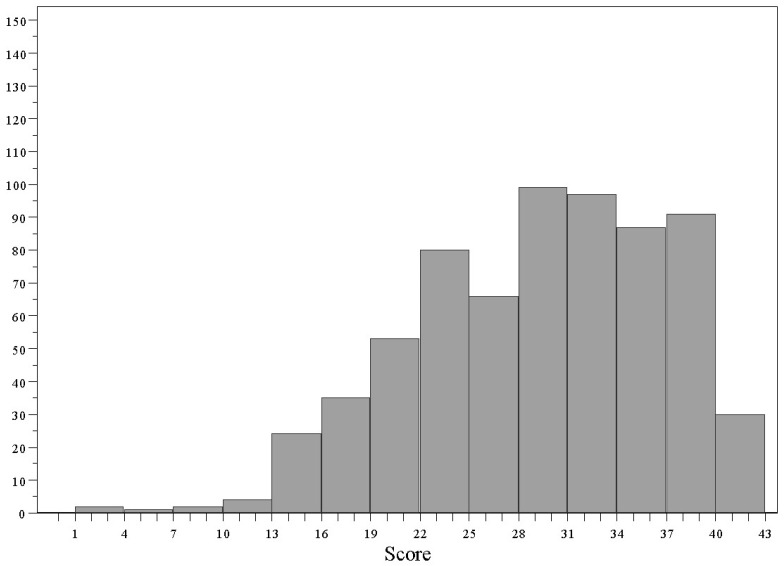
Distribution of total FKQ score. (Vertical axis: frequency).

**Table 1 nutrients-14-00691-t001:** Description of the sample.

Age in years, *mean (SD)*		33.9 (10.9)
Marital status, *n* (%)	Married	472 (70.3%)
	Single	106 (15.8%)
	Widowed	47 (7.0%)
	Partnered	32 (4.8%)
	Divorced	14 (2.1%)
Religion, *n* (%)	Christian	413 (61.6%)
	Muslim	258 (38.6%)
Education, *n* (%)	None	35 (5.2%)
	Primary schooling	331 (49.3%)
	Secondary schooling	200 (29.8%)
	High school	18 (2.7%)
	Tertiary schooling/technical	26 (3.9%)
	Diploma	36 (5.4%)
	Graduate	14 (2.1%)
	Postgraduate	11 (1.6%)
Nutrition-related qualification (yes), *n* (%)		16 (2.4%)
Work activity (yes)*, n* (%)		443 (66.0%)
Number of children, *median (IQR)*		2 (2)
Disabled persons in the family (yes), *n* (%)		30 (4.5%)
Monthly income in TZS (individual), *median (IQR)*		100,000 (150,000)
Monthly income in TZS (family), *median (IQR)*		250,000 (250,000)
Weekly expenditure for food in TZS, *median (IQR)*		30,000 (20,000)
Control of food expenditure, *n* (%)	Mother	445 (66.3%)
	Father	132 (19.7%)
	Children	13 (1.9%)
	Mother and father	76 (11.3%)
	Mother and children	4 (0.6%)
	Other	1 (0.2%)
Currently on a diet (yes)*, n* (%)		91 (13.6%)

**Table 2 nutrients-14-00691-t002:** Description of responses to nutrition items.

Section	Item	Wrong, *n* (%)	Not Sure, *n* (%)	Correct, *n* (%)	Note
B	1A	313 (46.7%)	45 (6.7%)	313 (46.7%)	
	1B	49 (7.3%)	14 (2.1%)	608 (90.6%)	DES+/MCA
	1C	88 (13.1%)	48 (7.2%)	535 (79.7%)	
	2	108 (16.1%)	77 (11.5%)	486 (72.4%)	
	3	78 (11.6%)	63 (9.4%)	530 (79.0%)	
	4A	118 (17.6%)	49 (7.3%)	504 (75.1%)	
	4B	212 (31.6%)	59 (8.8%)	400 (59.6%)	EFA
	4C	122 (18.2%)	105 (15.7%)	444 (66.2%)	
	4D	258 (38.5%)	43 (6.4%)	370 (55.1%)	EFA
	5A	221 (32.9%)	85 (12.7%)	365 (54.4%)	MCA
	5B	332 (49.5%)	126 (18.8%)	213 (31.7%)	EFA
	5C	145 (21.6%)	157 (23.4%)	369 (55.0%)	MCA
	5D	169 (25.2%)	144 (21.5%)	358 (53.4%)	MCA
C1	1A	478 (71.2%)	14 (2.1%)	179 (26.7%)	
	1B	48 (7.2%)	25 (3.7%)	598 (89.1%)	
	1C	168 (25.0%)	34 (5.1%)	469 (69.9%)	
	1D	131 (19.5%)	16 (2.4%)	524 (78.1%)	MCA
	1E	31 (4.6%)	6 (0.9%)	634 (94.5%)	DES+
	2A	279 (41.6%)	14 (2.1%)	378 (56.3%)	MCA
	2B	38 (5.7%)	21 (3.1%)	612 (91.2%)	DES+
	2C	300 (44.7%)	210 (31.3%)	161 (24.0%)	EFA
	2D	38 (5.7%)	115 (17.1%)	518 (77.2%)	
	2E	71 (10.6%)	37 (5.5%)	563 (83.9%)	
	2F	40 (6.0%)	20 (3.0%)	611 (91.1%)	DES+
	2G	46 (6.9%)	72 (10.7%)	553 (82.4%)	EFA
	2H	35 (5.2%)	75 (11.2%)	561 (83.6%)	EFA
	3A	104 (15.5%)	71 (10.6%)	496 (73.9%)	
	3B	65 (9.7%)	21 (3.1%)	585 (87.2%)	DES+
	3C	272 (40.5%)	66 (9.8%)	333 (49.6%)	
	3D	111 (16.5%)	67 (10.0%)	493 (73.5%)	
	3E	37 (5.5%)	23 (3.4%)	611 (91.1%)	DES+
	3F	159 (23.7%)	185 (27.6%)	327 (48.7%)	(MCA)
	4A	301 (44.9%)	237 (35.3%)	133 (19.8%)	MCA
	4B	135 (20.1%)	14 (2.1%)	522 (77.8%)	
	4C	170 (25.3%)	43 (6.4%)	458 (68.3%)	MCA
	4D	33 (4.9%)	17 (2.5%)	621 (92.6%)	DES+
	4E	22 (3.3%)	11 (1.6%)	638 (95.1%)	DES+
	4F	70 (10.4%)	98 (14.6%)	503 (75.0%)	
	4G	386 (57.5%)	20 (3.0%)	265 (39.5%)	MCA
	5A	142 (21.2%)	111 (16.5%)	418 (62.3%)	
	5B	64 (9.5%)	38 (5.7%)	569 (84.8%)	MCA
	5C	208 (31.0%)	43 (6.4%)	420 (62.6%)	
	5D	122 (18.2%)	93 (13.9%)	456 (68.0%)	
	5E	44 (6.6%)	27 (4.0%)	600 (49.4%)	DES+
	5F	290 (43.2%)	62 (9.2%)	319 (47.5%)	
	6A	21 (3.1%)	31 (4.6%)	619 (92.3%)	DES+
	6B	326 (48.6%)	36 (5.4%)	309 (46.1%)	MCA
	6C	23 (3.4%)	37 (5.5%)	611 (91.1%)	(DES+)
	6D	192 (28.6%)	35 (5.2%)	444 (66.2%)	
	6E	153 (22.8%)	45 (6.7%)	473 (70.5%)	MCA
	6F	46 (6.9%)	39 (5.8%)	586 (87.3%)	(DES+)
	6G	73 (10.9%)	35 (5.2%)	563 (83.9%)	
	6H	392 (58.4%)	36 (5.4%)	243 (36.2%)	EFA
	6I	138 (20.6%)	49 (7.3%)	484 (72.1%)	
	6L	74 (11.0%)	17 (2.5%)	580 (86.4%)	DES+
	7A	139 (20.7%)	60 (8.9%)	472 (70.3%)	
	7B	325 (48.4%)	183 (27.3%)	163 (24.3%)	EFA
	7C	142 (21.2%)	32 (4.8%)	497 (74.1%)	MCA
	7D	209 (31.2%)	47 (7.0%)	415 (61.9%)	
	7E	224 (33.4%)	75 (11.2%)	372 (55.4%)	MCA
	7F	79 (11.8%)	41 (6.1%)	551 (82.1%)	
C2	1	63 (9.4%)	18 (2.7%)	590 (87.9%)	(DES+)
	2	401 (59.8%)	74 (11.0%)	196 (29.2%)	EFA
	3	139 (20.7%)	105 (15.7%)	427 (63.6%)	
	4	365 (54.4%)	140 (20.9%)	166 (24.7%)	EFA
	5	109 (16.2%)	253 (37.7%)	309 (46.1%)	EFA
	6	259 (38.6%)	191 (28.5%)	221 (32.9%)	EFA
D	1A	83 (12.4%)	117 (17.4%)	471 (70.2%)	EFA
	1B	242 (36.1%)	111 (16.5%)	318 (47.4%)	
	1C	259 (38.6%)	184 (27.4%)	228 (34.0%)	
	2A	178 (26.5%)	30 (4.5%)	463 (69.0%)	MCA
	2B	6 (0.9%)	21 (3.1%)	644 (96.0%)	DES+
	2C	16 (2.4%)	227 (33.8%)	428 (63.8%)	EFA
	3A	135 (20.1%)	75 (11.2%)	461 (68.7%)	MCA
	3B	112 (16.7%)	200 (29.8%)	359 (53.5%)	
	3C	110 (16.4%)	127 (18.9%)	434 (64.7%)	
	4A	125 (18.6%)	152 (22.7%)	394 (85.7%)	
	4B	213 (31.7%)	149 (22.2%)	309 (46.1%)	
	4C	226 (33.7%)	147 (21.9%)	298 (44.4%)	
	4D	107 (16.0%)	99 (14.8%)	465 (69.3%)	
	4E	88 (13.1%)	102 (15.2%)	481 (71.7%)	
	5A	144 (21.5%)	104 (15.5%)	423 (63.0%)	
	5B	67 (10.0%)	74 (11.0%)	530 (79.0%)	
	5C	102 (15.2%)	68 (10.1%)	501 (74.7%)	
	6A	79 (11.8%)	66 (9.8%)	526 (78.4%)	EFA
	6B	80 (11.9%)	65 (9.7%)	526 (78.4%)	EFA
	6C	89 (13.3%)	72 (10.7%)	510 (76.0%)	EFA
	6D	84 (12.5%)	79 (11.8%)	508 (75.7%)	EFA

DES+: item with ≥90% correct answers. MCA: item showing an inconsistent scoring system in Multiple Correspondence Analysis (e.g., the score of the correct answer to a certain item was not consistent with the scores of the correct answers of other items in the same factor). EFA: item not integrated into the final factorial structure. All items with DES+, MCA, EFA were dropped from the final factorial structure unless DES+ or MCA are in parentheses.

**Table 3 nutrients-14-00691-t003:** Results of Exploratory Factor Analysis of the final structure of the questionnaire.

Factor	Item ID (Former)		Item ID (Final)		Factor Loading	Uniqueness
	*Section*	*Item*	*Section*	*Item*		
B (Explained Variance: 48.1%)	**B**	1A	B	1A	0.609	0.629
	B	1C	B	1B	0.641	0.589
	B	2	B	2B	0.685	0.531
	B	3	B	3B	0.776	0.398
	B	4A	B	4A	0.679	0.539
	B	4C	B	4B	0.755	0.430
C (Explained Variance: 55.2%)	C	1A	C	1A	0.676	0.543
	C	1B	C	1B	0.737	0.457
	C	1C	C	1C	0.810	0.344
D (Explained Variance: 61.5%)	C1	5A	D	1A	0.827	0.316
	C1	5C	D	1B	0.826	0.318
	C1	5D	D	1C	0.609	0.630
	C1	5F	D	1D	0.850	0.277
E (Explained Variance: 69.6%)	C1	6D	E	1A	0.818	0.332
	C1	6F	E	1B	0.728	0.469
	C1	6G	E	1C	0.900	0.190
	C1	6I	E	1D	0.881	0.225
F (Explained Variance: 50.5%)	D	3B	F	1A	0.804	0.654
	D	3C	F	1B	0.724	0.475
	C1	4B	F	2A	0.693	0.520
	C1	4F	F	2B	0.609	0.629
G (Explained Variance: 61.4%)	C1	3A	G	1A	0.755	0.430
	C1	3C	G	1B	0.774	0.401
	C1	3D	G	1C	0.843	0.290
	C1	7A	G	2A	0.799	0.362
	C1	7D	G	2B	0.725	0.475
	C1	7F	G	2C	0.799	0.362
H (Explained Variance: 54.8%)	C1	2D	H	1A	0.630	0.603
	C1	2E	H	1B	0.833	0.307
	C1	6C	H	2A	0.745	0.445
I (Explained Variance: 45.6%)	C2	1	I	1	0.753	0.434
	C2	3	I	2	0.751	0.436
	C1	3F	I	3A	0.488	0.762
L (Explained Variance: 59.1%)	D	1B	L	1A	0.655	0.571
	D	1C	L	1B	0.732	0.464
	D	4A	L	1C	0.848	0.281
	D	4B	L	1D	0.758	0.426
	D	4C	L	1E	0.785	0.383
	D	4D	L	2A	0.877	0.231
	D	4E	L	2B	0.687	0.528
	D	5A	L	2C	0.780	0.392
	D	5B	L		0.718	0.485
	D	5C	L		0.817	0.332

## Data Availability

Data are contained within the article or Appendix A.

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
