# Peer review of "Validation of a Food Knowledge Questionnaire on Tanzanian Women of Childbearing Age"

_nutrients, 2022, doi:10.3390/nu14030691_

Round 1

Reviewer 1 Report

The manuscript represents a comprehensive study aimed at verifying the accuracy of using this questionnaire regarding food knowledge on Tanzanian women of child bearing age (CBA) and or women in similar socio-cultural environment.

It has shown that the FKQ as adapted for the Tanzanian context has good construct validity and content validity to assess the knowledge and food practices among Tanzanian (CBA) women and can be applied in other studies to identify women prone to unhealthy nutritional status.

This appears to be the first and only study that has provided evidence of FK assessment in Tanzanian CBA women, its relatively easy to use for similar public health interventions and is fairly reliable given the sample size involved in the study. It is also fairly rapid to administer. There are however, some weakness pointed out including omission of water intake in the version applied and the need for further evidence to support the validity of the proposed total FKQ score.

This is a sound piece of research and provides sufficient details of the subject matter in addition, the manuscript is well written. I recommend that it should be accepted for publication in the present form.

Author Response

The authors would like to thank Reviewer 1 for the comment and appreciation of their research work presented in this manuscript.

Reviewer 2 Report

General Comment:  This is a very well-written manuscript on the validity of a food knowledge questionnaire for a specific group – childbearing Tanzanian women.  Authors have organized and described the results clearly.  Evaluating the “validity” of the questionnaire provides a model for future studies and is indeed the strength of this study.  I completely agree that the confirmation of the validity of these types of surveys and using them to test the knowledge of nutrition factors will be of great benefit to health care professionals.    

  1. Construct and content validity are complex issues. The authors have described a comprehensive approach in the methodology regarding the evaluation of these concepts, especially the former.  Can it be assumed that construct validity of this questionnaire is reproducible in diverse groups, e.g., men, different nationalities and cultures?    
  2. I’m not sure if many readers will be familiar with the term “trained enumerator”. How does one become a trained enumerator?  Will a trained enumerator be required to administer the questionnaire in future studies or is it possible to self-administer the questionnaire?
  3. The women’s education level affected the factor’s score, and this is certainly expected. I am curious if age showed similar results.  Did the youngest group show lower scores? 
  4. If a traditional or cultural diet of Tanzania exists, I suggest a comment be added in the Introduction or Discussion on such a diet and if it has changed throughout history. Additionally, how prevalent is malnutrition (line 340) in Tanzania and how has this changed?  Can comments be added regarding current health conditions of the Tanzanian women, e.g., hypertension, diabetes, obesity, etc?

Author Response

The authors thank reviewers 2 for the comments and they reply point by point in the attached file. Please see the attachment.

Reviewer 3 Report

First of all, thank you for reviewing this manuscript which deals with the validation of a survey in a well-defined population and geographical context.
It is a validation of a well-targeted survey. The introduction makes clear the subject matter, justifies the development of the research, and provides an adequate amount of bibliography.
The methodology and results are consistent and clearly presented, explaining the results obtained.
In the discussion, it would be appreciated if the results could be compared with some published articles that deal with the subject in a similar as well as opposite way. This part of the discussion seems to be somewhat lacking.
I would add a section on limitations outside the discussion.
I would also like to end this section with some initiatives in the form of "solutions to the issue".
For the conclusion section I would do it separately from the discussions.
Good research and I hope that the article will improve with the indications provided.

Author Response

The authors thank reviewers 3 for the comments and they reply point by point in the attached file. Please see the attachment.
